# Immunomagnetic-Enriched Subpopulations of Melanoma Circulating Tumour Cells (CTCs) Exhibit Distinct Transcriptome Profiles

**DOI:** 10.3390/cancers11020157

**Published:** 2019-01-30

**Authors:** Carlos Aya-Bonilla, Elin S. Gray, Jayapal Manikandan, James B. Freeman, Pauline Zaenker, Anna L. Reid, Muhammad A. Khattak, Markus H. Frank, Michael Millward, Mel Ziman

**Affiliations:** 1School of Medical and Health Sciences, Edith Cowan University, Perth, WA 6027, Australia; c.ayabonilla@ecu.edu.au (C.A.-B.); jamie_nocturne@hotmail.com (J.B.F.); paulinez@our.ecu.edu.au (P.Z.); a.reid@ecu.edu.au (A.L.R.); Markus.Frank@childrens.harvard.edu (M.H.F.); m.ziman@ecu.edu.au (M.Z.); 2NanoString, Inc., Seattle, WA 98109, USA; kanmani.j@gmail.com; 3School of Medicine, University of Western Australia, Crawley, WA 6009, Australia; adnan.khattak@health.wa.gov.au (M.A.K.); michael.millward@uwa.edu.au (M.M.); 4Department of Medical Oncology, Sir Charles Gairdner Hospital, Nedlands, WA 6009, Australia; 5Transplantation Research Program, Boston Children’s Hospital and Department of Dermatology, Brigham and Women’s Hospital, Harvard Medical School, Boston, MA 02115, USA; 6Harvard Stem Cell Institute, Harvard University, Cambridge, MA 02138, USA; 7School of Biomedical Science, University of Western Australia, Crawley, WA 6009, Australia

**Keywords:** circulating tumour cells, melanoma, ABCB5, MCSP, gene expression

## Abstract

Cutaneous melanoma circulating tumour cells (CTCs) are phenotypically and molecularly heterogeneous. We profiled the gene expression of CTC subpopulations immunomagnetic-captured by targeting either the melanoma-associated marker, MCSP, or the melanoma-initiating marker, ABCB5. Firstly, the expression of a subset of melanoma genes was investigated by RT-PCR in MCSP-enriched and ABCB5-enriched CTCs isolated from a total of 59 blood draws from 39 melanoma cases. Of these, 6 MCSP- and 6 ABCB5-enriched CTC fractions were further analysed using a genome-wide gene expression microarray. The transcriptional programs of both CTC subtypes included cell survival maintenance, cell proliferation, and migration pathways. ABCB5-enriched CTCs were specifically characterised by up-regulation of genes involved in epithelial to mesenchymal transition (EMT), suggesting an invasive phenotype. These findings underscore the presence of at least two distinct melanoma CTC subpopulations with distinct transcriptional programs, which may have distinct roles in disease progression and response to therapy.

## 1. Background

Metastatic melanoma is an aggressive tumour that is extremely difficult to treat, thought due to the heterogenous nature of the tumours [1,2]. The isolation and study of circulating tumour cells (CTCs) is an emerging field of basic and clinical cancer research worldwide. CTCs, released by all the concurrent tumours within the patient, are responsible for the haematogenous spread of the cancer. CTCs offer an accessible source to ascertain the biology of all tumours within the patient in real time and provide a biomarker to anticipate clinical outcome and monitor treatment response in a personalised manner [3,4], and perhaps a readout of the heterogeneity of melanoma tumours.

In melanoma, CTCs have to date been shown to provide prognostic information and evidence of treatment response in real time [5,6,7,8,9]. Melanoma CTCs are also known to be highly heterogeneous [5,10,11,12] and to show differential response to treatment [5,8]. In fact, we have shown that subpopulations of CTCs expressing the receptor activator of nuclear factor kappa B (RANK) and the programmed death-ligand 1 (PD-L1) are associated with response to MAPK and PD-1 inhibitory treatments, respectively [5,6]. More detailed phenotypic and molecular characterisation of CTCs is hindered by the lack of a ubiquitous marker for CTC detection, as reviewed by Marsavela, et al. [13]. CTCs derived from epithelial cancers commonly express cytokeratin and/or EpCAM (epithelial cell adhesion molecule) [14,15,16], which are used for CTC isolation and detection. By contrast, melanoma CTCs do not commonly express these markers [17,18], since melanocytes originate from the neural crest lineage [19,20]. Therefore, the isolation and detection of melanoma CTCs has to date relied on the use of a variety of markers, not well formalised across the literature [5,9,10,11,12,21].

Conventionally, CTCs have been captured or detected by targeting the melanoma-associated chondroitin sulphate proteoglycan (MCSP, HMW-MAA, CSPG4, NG2) [5,7,8,9,10,22,23,24]. This cell surface proteoglycan, reported to be overexpressed in over 90% of melanoma tissue samples, is thought to increase the tumorigenic potential of melanoma cells, by increasing cell proliferation, survival, invasion, spreading, and migration [25,26]. However, we found in a previous study [5] that CTCs more commonly express melanoma-initiating markers, such as the ATP-binding cassette sub-family B member 5 (ABCB5) than melanoma-associated markers, such as MCSP. The transmembrane transporter ABCB5 is associated with melanomagenesis, stem cell maintenance, metastasis, and treatment resistance to MAPK inhibition [27,28,29]. Interestingly, we showed that ABCB5 is expressed only by rare subpopulations within patient-matched tumour tissues [5], supporting the hypothesis that CTCs are derived from rare subpopulations in the tumour with the potential to initiate metastases [28,30]. Taken together, it seems clear that cells positive for MCSP and ABCB5 represent two different CTC subpopulations, yet there is no knowledge about the differential biology of these subpopulations.

To better understand the molecular diversity of CTC subpopulations and their role in melanoma metastasis, we profiled the transcriptome of CTCs enriched immunomagnetically by targeting MCSP or ABCB5, using firstly an RT-PCR assay involving a panel of five melanoma-specific genes, then microarray genome-wide expression profiling. These approaches firstly identified the presence of CTCs in the samples (five specific genes), then, using microarrays, we identified differentially expressed genes and pathways in these two CTC subpopulations. Finally, we compared the transcriptomes of these enriched cells to publicly available single cell RNAseq data from melanoma tumours [31], to better understand the features of melanoma tumour cells that give rise to CTCs.

## 2. Results

### 2.1. Expression of Five Melanoma-Specific Transcripts in Immunomagnetically Enriched ABCB5 and MCSP CTC Fractions from Metastatic Melanoma Patients

Fifty-nine blood samples from 39 metastatic melanoma patients were subjected to CTC enrichment using immunomagnetic beads conjugated to anti-MCSP or anti-ABCB5 antibodies (Table 1).

Total RNA was isolated from the CTC fractions and molecularly characterised by assessing the expression of a panel of five melanoma-specific genes by a sensitive RT-PCR assay. As previously reported [12,32], this assay assessed the gene expression of melanoma specific genes, *MLANA/MART-1* (melanoma antigen recognized by T cells), *TYR* (tyrosinase), *MAGEA3* (melanoma antigen family A3), *PAX3* (paired box protein Pax-3 isoform 3), and *ABCB5* at a sensitivity level of at least one melanoma cell in a background of 1 × 10^5^ WBCs. A total of 25 (42%) samples were found positive for CTCs based on expression of any one of these transcripts in either of the CTC fractions enriched from their blood. Of those, nine patients had detectable transcripts in both fractions, six were only positive in the MCSP-enriched fraction, and 10 were only positive in the ABCB5-enriched fraction. This suggests that CTCs can be isolated by targeting either MCSP or ABCB5 cell surface markers.

As illustrated in Figure 1, differential expression of the melanoma specific transcripts was observed between and within CTC fractions enriched with MCSP- or ABCB5-coated beads, suggesting the molecular heterogeneity of these two CTC subpopulations. In detail, MCSP-enriched CTCs were characterised by a higher frequency of *MLANA*, *TYR*, and *PAX3* and the exclusive expression of *MAGEA3* relative to ABCB5-enriched CTCs. On the other hand, the expression of *ABCB5* was most commonly detected in ABCB5-enriched CTC fraction, but no other genes were co-detected with ABCB5. Conversely, in MCSP-enriched CTC fractions, ABCB5 transcripts were co-detected with *TYR*, *PAX3*, *MLANA* or *MAGEA3* transcripts in three CTC fractions, whilst five CTC fractions had exclusive ABCB5 expression, and the remaining expressed other melanocyte lineage genes (Figure 1). No expression of these five genes was detected in MCSP- or ABCB5-enriched fractions derived from similarly evaluated healthy donors (*N* = 16).

### 2.2. MCSP and ABCB5 CTC Subpopulations Exhibit Distinct Gene Expression Profiles

To further characterise the MCSP and ABCB5 CTCs, total RNA isolated from the dichotomised CTC enriched fractions of six MCSP-enriched CTC samples and six ABCB5-enriched CTC samples identified as RT-PCR positive for any of the five melanoma specific genes was further characterised by whole genome expression microarray analysis. CTC samples were profiled and contrasted to the same number of samples from six healthy control bloods enriched with either MCSP or ABCB5 coated beads (Appendix A). Gene expression analysis identified 308 genes that were differentially expressed between MCSP-enriched CTC fractions from metastatic melanoma patients and healthy controls. From those, 260 and 48 genes were significantly up- or down-regulated, respectively (Appendix A). For ABCB5, 210 genes were differentially expressed in CTC fractions derived from melanoma patients relative to those from controls, with 187 genes significantly up-regulated and 23 genes significantly down-regulated (Appendix A).

An unsupervised hierarchical clustering analysis successfully differentiated patient immunomagnetic-enriched fractions from those of healthy control samples. Additionally, this analysis partially clustered the enriched CTC fractions based on whether these were enriched using MCSP or ABCB5 coated beads; two ABCB5-CTC fractions were the exception exhibiting very distinct gene expression patterns (Figure 2A).

Comparison of the genes significantly up-regulated in MCSP- and ABCB5-enriched patient fractions revealed differential gene expression patterns, with only 15 genes commonly up-regulated by both MCSP and ABCB5 CTC fractions (Figure 2B and Appendix A). A total of 245 genes were only up-regulated in MCSP CTC fractions (Appendix A) and 172 genes were only up-regulated in ABCB5 CTC fractions (Appendix A) relative to healthy controls, suggesting that MCSP and ABCB5 CTCs might be two molecularly different cell types.

### 2.3. Tumour Necrosis Factor Alpha (TNFA) Signalling and Epithelial Mesenchymal Transition (EMT) Are Significantly Enriched in the MCSP and ABCB5 CTC Fractions, Respectively

To gain a better understanding of the role of these up-regulated genes in CTC subpopulations, we performed gene set enrichment analyses (GSEA) of the up-regulated genes in the MCSP- or ABCB5-enriched CTC fractions. This approach identified 10 distinctive hallmarks in each CTC subtype, with the TNFα signalling being the most prominent in the MCSP-enriched fraction, whereas the EMT hallmark was the most prominent in the ABCB5- fraction (Table 2 and Table 3). Of note, the KRAS signalling and the PI3K-AKT/mT ORC1 signalling hallmarks were found to be commonly enriched between the MCSP- and ABCB5-enriched CTC fractions. However, the genes enriching these cellular hallmarks differ for each CTC type. Specifically, M CSP CTCs were characterised by the enrichment of the TNFA/NFKB, IL2/STAT5 and IL6/JAK/STAT3 signalling hallmarks. On the other hand, ABCB5 CTCs were characterised by enrichment of the apical junction and EMT hallmarks (Figure 3).

### 2.4. In Silico Analysis Identifies Melanoma Cell-Associated Genes in the Gene Expression Profiles of the MCSP- and ABCB5-Enriched CTC Fractions

To discern which of the genes found to be up-regulated in the MCSP- and ABCB5-enriched CTC fractions were also preferentially expressed by single melanoma tumour cells, an in silico analytical approach was performed using publicly available single cell RNAseq data from melanoma tumours (GSE72056) [31]. Comparison of gene expression profiles of 1257 single melanoma cells and 3256 single nonmalignant cells from these tissues identified a total of 847 genes which were more specifically expressed by single melanoma cells than by the nonmalignant cells (Figure 4 and Appendix A). Subsequently, this list of melanoma-expressed genes was compared against the up-regulated genes in the MCSP- and ABCB5-enriched CTC fractions.

This approach identified a total of 11 and 10 genes (Table 4) expressed in single tumour cells that were overexpressed in the MCSP- and ABCB5-enriched CTC fractions, respectively. Of note, *WIPI1* (WD repeat domain phosphoinositide-interacting protein 1) and *DDIT3* (DNA damage inducible transcript 3) were the only two genes found to be overexpressed in both CTC fractions. Additionally, a total of 196 genes were also identified to be specifically expressed by the nonmalignant cells. However, only two of those genes were also found to be up-regulated in either MCSP- or ABCB5-enriched CTC fractions.

## 3. Discussion

To improve cancer therapy, it is critical to target metastasizing cells. A better understanding of the biology of CTCs and the molecular mechanisms dictating melanoma cell spreading offers a potential avenue to inform treatment. Moreover, by understanding the biological heterogeneity of CTCs, methods to identify CTC subpopulations that seed metastases and/or provide resistance to treatment can be developed. Here, in this study, we report on CTC subpopulations expressing either MCSP, a melanoma-associated marker [25], or ABCB5, a melanoma-initiating marker [28], and show that they display distinct transcriptomic profiles. This study confirms the phenotypic and molecular heterogeneity observed in melanoma CTCs [5,9,10,11,12,33] and highlights genes and cellular pathways that may be associated with the biology of MCSP or ABCB5 CTC subtypes.

The most significant finding from this study is that, based on gene expression data, MCSP- and ABCB5-enriched CTCs are two distinct cell subpopulations. This is based on the limited overlap observed between the genes found to be up-regulated in MCSP and ABCB5 CTCs and the different pathways prominently enriched in these two CTC-enriched fractions. MCSP is a melanocytic membrane proteoglycan involved in melanomagenesis, tumour growth, motility, and tissue invasion of melanoma cells through activation of integrin function, FAK (focal adhesion kinase) signalling, ERK signalling, and matrix metalloproteinase 2 [25,34,35,36,37]. Because MCSP is expressed in over 90% of melanomas [25], this cell surface proteoglycan has been used as a conventional marker for the capture and detection of CTCs, but little is known of its efficacy in identifying CTCs or about the biology of MCSP-positive CTCs [5,7,8,9,10,11,22,23,24]. Here, our gene set enrichment analysis (GSEA) performed with genes found to be significantly up-regulated in the MCSP-enriched CTC fractions, showed that the TNFA_SIGNALING_VIA_NFKB hallmark was most significantly enriched in this CTC-subtype. Furthermore, IL2_STAT5_SIGNALING, MTORC1_SIGNALING, and IL6_JAK_STAT3_SIGNALING were hallmarks of the MCSP-enriched CTC fractions. The enrichment of these cellular networks in these CTC fractions is interesting as MCSP tumorigenic signalling depends on the activation of SRC kinases, FAK, STAT transcription factors and apoptotic regulators including TP53, TOR and JNK [26]. These findings indicate that MCSP CTCs are equipped with tumorigenic capabilities, such as maintenance of cell survival, cell proliferation, and migration, as they travel through the bloodstream (Figure 3). Nevertheless, the limited overlap in gene expression observed between single tumour cells and these CTC fractions might indicate that these cellular networks might function differently in CTCs in comparison to tissue-derived tumour cells. An in-depth understanding of MCSP-driven signalling in the biology of MCSP-positive tumour cells and CTCs and its role in melanomagenesis and metastasis demands further investigation.

Contrary to MCSP-enriched CTC fractions, ABCB5-enriched CTC fractions showed a significant enrichment of the epithelial mesenchymal transition and the apical junction hallmarks, which are crucial steps in metastasis [38]. ABCB5, an ATP-binding cassette, is a melanoma ‘stem cell’ marker of a slow-cycling population of tumour cells with self-renewal, proliferation, differentiation, and tumorigenicity capabilities [28,29,30]. Moreover, melanoma tumour cells expressing ABCB5 have been found to escape chemotherapy and MAPK inhibition, possibly as a result of its drug-efflux function [27,28]. The significant enrichment of EMT genes in ABCB5-enriched CTCs is interesting since EMT is a process in melanoma by which the melanoma cells lose their melanocytic phenotype and acquire a more invasive phenotype [39]. This is particularly important as, except for *CALU*, the remaining seven EMT genes found to be up-regulated in ABCB5-enriched CTC fractions have been previously associated with melanoma metastasis, invasion, and/or tumour growth. Previously, ABCB5 expression has been associated with melanoma progression and resistance [28], yet it is unclear how ABCB5 drives melanoma spreading. Recently, Yao, et al. [40] found that ABCB5 promotes metastasis and invasion in breast cancer by up-regulating ZEB1 (zinc finger e-box binding homeobox 1), an EMT-driving transcription factor that is also involved in melanoma EMT [39]. The significant enrichment of the apical junction hallmark in ABCB5 CTCs is interesting as loss of cell polarity represents a step towards EMT cancer [38]. It is interesting that enrichment of EMT and apical junction hallmarks is found specifically in ABCB5 CTCs and suggests that both cellular processes may be key in ABCB5 CTC biology, since ABCB5 indirectly activates WNT-IL8 signalling by negative regulation of the WNT-repressor, WFDC1 [29]. Thus, ABCB5 CTCs might be enriched for EMT and apical junction transcriptional programs (Figure 3B) via WNT signalling, which is known to regulate both programs. Additionally, the significant enrichment of melanoma EMT-involved pathways, such as the MTORC1_SIGNALING and PI3K_AKT_MTOR_SIGNALING hallmarks in these CTC fractions, suggests an invasive transcriptional programming of ABCB5 CTCs [39] (Figure 3B). The finding of PI3K/AKT in ABCB5 CTCs is interesting as the PI3K-AKT signalling pathway has been shown to regulate the quiescence of melanoma stem-cells characterised by high ABCB5 expression [41]. The presence of an EMT signature in the ABCB5 CTC fractions indicates that activation of this signature might play a critical role either in the migration of these CTCs from the tumour or in their ability to seed new tumours. Since ABCB5 is a common marker in CTCs but expressed only in rare cell subpopulations in matched melanoma tumour tissues [5], it is possible that rare tumour subpopulations, through ABCB5 expression, might acquire invasive capabilities via EMT activation.

In this study, we observed that MCSP and ABCB5 CTC fractions had 11 and 10 genes commonly expressed with single melanoma tumour cells, respectively. The limited overlap between gene expression profiles of single tumour cells and both CTC fractions might reflect differences in the transcriptional programming of CTCs and tumours. Interestingly, only *WIPI1* and *DDIT3* were shared between both CTC fractions and single melanoma cells. *WIPI1* has been involved in melanosome formation and regulation of melanogenic enzymes, such as MITF and its targets, via negative regulation of TORC1 signalling [42]. Additionally, *WIPI1* was identified as the signature gene able to differentiate primary melanomas from normal skin [43]. However, we observed upregulation of WIPI1 as well as of genes involved in mTORC1 signalling in both fractions, with no evidence of MITF transcriptional programming. Altogether, this suggests dysregulation of these signalling pathways in melanoma and CTC biology. Thus, further investigation on the role of WIPI1 and mTORC1 upregulation and signalling in CTC biology is required. On the other hand, *DDIT3,* a mediator of endoplasmic reticulum stress-mediated apoptosis [44], has not been implicated in melanoma biology to date. However, DDIT3 was up-regulated as a result of the apoptotic activity triggered by the compound HA15 in melanoma cell lines [45], as was SEC31A, [45] a gene found up-regulated in MCSP CTC fractions, Altogether, the role of *WIPI1* and *DDIT3* may indicate common molecular mechanisms in CTC biology. It is critical to understand such mechanisms in order to determine the changes in transcriptional programming that tumour cells undergo to become CTCs.

With regard to the genes specifically up-regulated in the MCSP CTC fraction, the majority have been involved in metastasis (*LOXL4, ST6GALNAC2, PYGL)*, tumour growth (*PLK2*, *CYSTM1, GLUL*), and/or melanoma biology (*SEC31A, WIPI1*, *SKP1*) [42,43,45,46,47,48,49,50,51,52,53]. Similarly, the up-regulated genes in the ABCB5 CTC fractions were involved in metastasis (*CALU*, *CYB5R1, DUSP3, CREG1, ENDOD1*), tumour growth, and/or melanoma biology (*H1F0, SCNA, WIPI1, TXN2*) [42,43,54,55,56,57,58,59,60]. These findings support the suggestion that these two CTC subpopulations exhibit different gene expression patterns but similar molecular mechanisms underlying their biology.

By interrogating the transcriptome of MCSP- and ABCB5-enriched CTCs, we herein provide novel evidence supporting the transcriptional heterogeneity within and amongst CTC subpopulations [5,10,11,12,24,33]. In addition, this study provides an insight into the molecular mechanisms underlying the biology of MCSP and ABCB5 CTCs. However, based on the low detection rates of CTCs, described here (42%) and elsewhere (40–87%) [5,7,9,11,12,21,33], it is likely that other CTC subpopulations currently remain undetected. Unbiased isolation and detection methods are required in order to capture the true heterogeneity of CTCs. Such studies will provide CTC subtype specific markers other than MCSP and ABCB5 and will provide a better understanding of the biology of specific CTC subtypes as well as their association with melanoma biology. In addition, analysis of pure CTCs fractions or of single CTCs is needed to ascertain their transcriptional programming without being confounded by the background of white blood cells.

## 4. Methods

### 4.1. Patient Recruitment and Blood Collection

Metastatic melanoma patients were enrolled in the study at Sir Charles Gardner Hospital and Royal Perth Hospital in Perth, Western Australia. Written informed consent was obtained from all patients. The study was approved by the Human Research Ethics Committees of Edith Cowan University (No. 11543) and Sir Charles Gairdner Hospital (No. 2013-246). Peripheral blood samples from healthy volunteers and patients were collected by phlebotomists into 4 mL BD Vacutainer^®^ K2 EDTA tubes (BD Life Sciences, Franklin Lakes, NJ, USA), from which the first few millilitres were discarded to avoid epithelial contamination, then refrigerated at 4 °C and analysed within 24 h.

### 4.2. CTC Enrichment and Nucleic Acid Extraction

CTCs were enriched as previously described [10]. In summary, whole blood was treated with red blood cell lysing buffer and the remaining cells were incubated with immunomagnetic beads coated with antibodies against MCSP (clone 9.2.27, BD Biosciences, Franklin Lakes, NJ, USA) or ABCB5 (clone 3C2-1D12) [28]. After washing, RNA was extracted from these CTC fractions using previously described methodologies [12,32].

### 4.3. RT-PCR

Extracted RNA (6 µL) was used for cDNA preparation using the SuperScript^®^ VILO™ cDNA Synthesis Kit (Thermo Fisher Scientific, Waltham, MA, USA) followed by a TaqMan^®^ PreAmp Master Mix Kit (Thermo Fisher Scientific). The presence of transcripts for melanoma genes *MLANA*, *TYR*, *MAGEA3*, *PAX3*, and *ABCB5* was determined using TaqMan^®^ probes in a ViiA 7 Real-time instrument (Thermo Fisher Scientific). Detection of 18S rRNA was used as an endogenous control gene [12,32].

### 4.4. Microarray

RNA quantity and quality were analysed with a Pico Bioanalyzer chip (Agilent Technologies, Santa Clara, CA, USA). Samples were prepared using the NuGEN Pico WTA kit (NuGEN Technologies, San Carlos, CA, USA). Labelling and hybridization according to the Affymetrix^®^ GeneChip^®^ genome 2.0 ST array protocol (Affymetrix, Santa Clara, CA, USA) was performed at the Ramaciotti Centre for Gene Function Analysis (University of New South Wales, Sydney, Australia).

### 4.5. Microarray Data Analysis

Microarray data were analysed using Partek Genomics Suite software version 6.5 (Partek Inc., St. Louis, MO, USA). Raw data were normalised using the robust multi-array average (RMA) normalisation approach. The data fulfil the MIAME (Minimum Information About a Microarray Experiment) guidelines and have been uploaded into the National Center for Biotechnology Information (NCBI)’s GEO database (accession number GSE113166). The differentially expressed gene list was generated with ANOVA *p* < 0.05 and fold difference 2 vs. respective controls in either cell type (MSCP or ABCB5 CTCs). Unsupervised two-dimensional hierarchical clustering with complete linkage was performed on differentially expressed genes using Spearman’s correlation as a similarity matrix.

### 4.6. Gene Set Enrichment Analysis (GSEA)

Genes found to be significantly up-regulated (≥2-fold change and *p* < 0.05) in the MCSP-enriched and ABCB5-enriched CTC fractions were contrasted with collections from the Molecular Signatures Database (MSigDB v6.0; Broad Institute, Boston, Massachusetts, USA) [61]. Particularly, we used the Hallmark gene set collection in order to reduce redundancy of the pathways being enriched and obtain more biologically meaningful results [62]. For each CTC fraction, the top ten hallmark gene sets with a *p*-value below 0.01 and a false discovery rate (FDR) *q*-value below 0.01 were obtained and analysed.

### 4.7. Identification of Tumour-Derived Genes in the MCSP- and ABCB5-Enriched Fractions

To identify genes that are commonly expressed between melanoma tumours and the MCSP- and ABCB5-enriched CTC fractions, we used publicly available single cell RNAseq data (Gene Expression Omnibus series no. GSE72056) from melanoma tumours. This dataset comprises RNAseq data of 1257 single tumour cells and 3256 single nonmalignant cells from a total of 19 melanoma tumours [31]. Visualisation and analysis of this single cell RNAseq dataset were performed on the SeqGeq™ platform (v1.0.1; FlowJo^®^, LLC, Ashland, Oregon). Firstly, genes that were more specifically expressed by melanoma tumour cells than by nonmalignant cells were identified by comparing the gene expression profiles of all single tumour cells and nonmalignant cells followed by gating them out in a pivot plot. In this plot, the sum of the expression levels for each gene of a population of interest is normalised to the maximum each gene can be expressed in the comparative population. Thus, a gene that is closer to an axis is more expressed by the population the axis belongs to. Secondly, these melanoma-specific genes were contrasted to the list of genes found to be up-regulated in the MCSP- and ABCB5-enriched CTC fractions in order to detect commonly expressed genes between melanoma tumour cells and CTC fractions.

## 5. Conclusions

Overall, gene expression profiling of MCSP-enriched and ABCB5-enriched CTCs provide evidence that MCSP- and ABCB5-enriched CTCs are two transcriptionally distinct CTC subpopulations with MCSP CTCs, the more melanocytic cell subpopulation, whilst ABCB5 CTCs may be a more invasive cell subpopulation as a result of their EMT profiles. This study is the first to characterise two different melanoma CTC subpopulations, unveiling the differences in their transcriptional programs and suggesting distinct biological pathways activated in these cells. These findings will lead to further studies to determine the biology and role of these CTC subpopulations in melanoma metastasis.

## Figures and Tables

**Figure 1 cancers-11-00157-f001:**
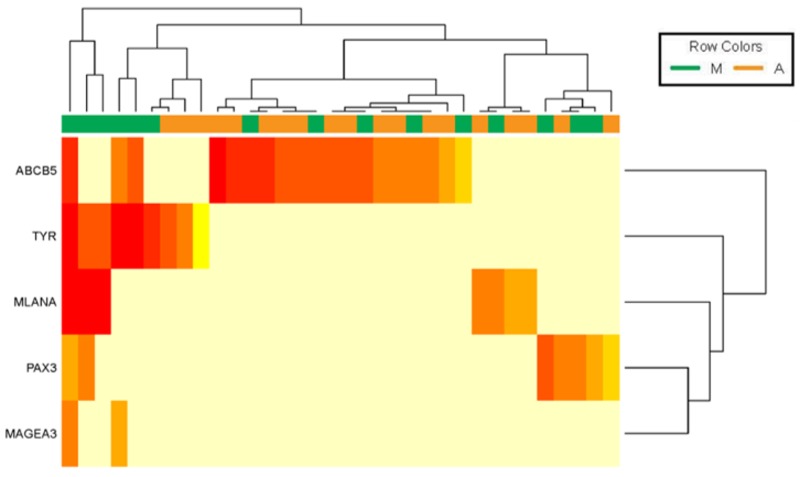
Heterogeneity in gene expression of MCSP- and ABCB5-enriched circulating tumour cell (CTC) fractions. Gene expression profiling of five melanoma-specific genes in CTC fractions enriched with MCSP- (**M**) and ABCB5- (**A**) coated beads that were positive for any transcript (25 out of 59 samples). Heatmap represents the expression levels of the melanoma-associated genes *MLANA, TYR, MAGEA, ABCB5*, and *PAX3*, which were assessed by qRT-PCR. All samples were positive for 18S rRNA (not shown), used as endogenous control. Each heatmap square corresponds to a Ct value for a target gene and a given sample; a higher tone of red indicates a higher gene expression and light yellow indicates no transcripts detected. Unsupervised hierarchical clustering demonstrates the inter- and intra- molecular heterogeneity of CTC subpopulations.

**Figure 2 cancers-11-00157-f002:**
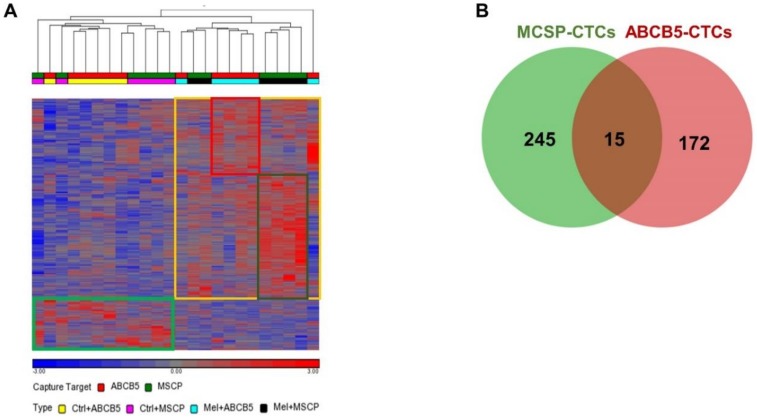
MCSP and ABCB5 CTCs have distinctive gene expression patterns. (**A**) Unsupervised hierarchical clustering analysis of differentially expressed genes between CTC fractions enriched from samples of healthy donors (*N* = 6) and melanoma patients (*N* = 6) using immunomagnetic beads targeting MCSP or ABCB5. Each row is a single gene and each column is a single sample. Red indicated up-regulation and blue indicates down-regulation according to the colour scale at the bottom. The squares indicate the genes that were up-regulated in melanomas (yellow box) and in controls (light green box). Genes that are differentially expressed (with ANOVA *p* < 0.05 and fold difference 2 vs. respective controls) in either cell type (MSCP or ABCB5 CTCs) were used to perform an unsupervised hierarchical clustering. This analytical approach discriminated the gene expression patterns from MCSP- (dark green box) and ABCB5-enriched (red box) CTC fractions. (**B**) Comparison of up-regulated genes between MCSP- (green circle) and ABCB5-enriched (red circle) CTC fractions found only 15 genes shared between these two CTC fractions.

**Figure 3 cancers-11-00157-f003:**
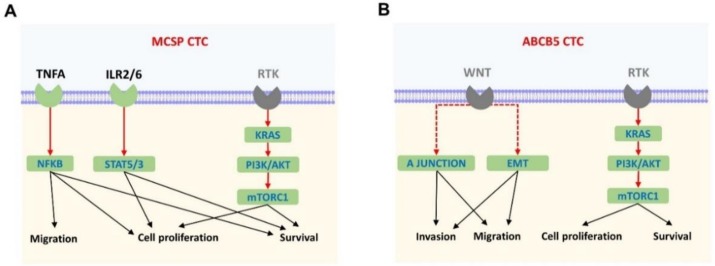
MCSP- and ABCB5-enriched CTCs exhibit distinct transcriptional programming, with ABCB5 CTCs displaying an invasive phenotype. Schematic representation of cellular hallmarks enriched by the genes found to be differentially up-regulated in (**A**) MCSP- and (**B**) ABCB5-enriched CTCs. Enriched cellular hallmarks in CTCs are shown in green boxes, while genes shown in grey are those that were not found enriched in this study but are known to be important for melanomagenesis. Canonical pathway signalling is indicated by continuous arrows and dotted arrows indicates a previously reported interaction.

**Figure 4 cancers-11-00157-f004:**
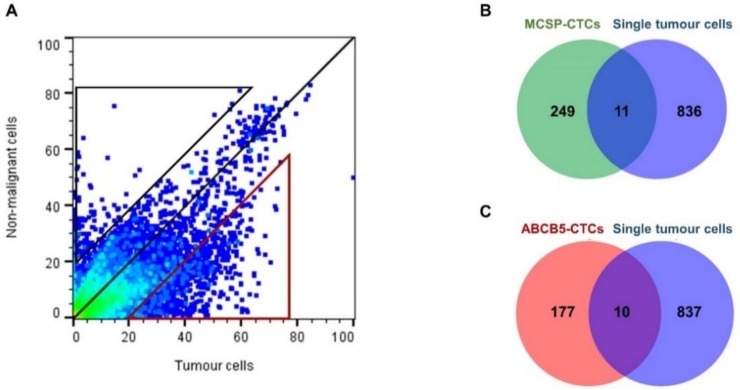
Identification of genes specific to single melanoma tumour cells that are also up-regulated in MCSP- and ABCB5-enriched CTC fractions. (**A**) Pivot plot comparing gene expression profiles of single tumour cells and nonmalignant cells, from publicly available data [31], identified 847 genes that are specifically expressed by single tumour cells (red triangle) and 196 genes specifically expressed by single nonmalignant cells (black triangle). Comparison of these tumour-specific genes against the genes found to be up-regulated in MCSP (**B**) and ABCB5 CTC fractions (**C**) identified, respectively, 11 and 10 genes that are commonly expressed by single melanoma tumours as well as by MCSP or ABCB5 CTC fractions.

**Table 1 cancers-11-00157-t001:** Demographic and clinical information of metastatic melanoma patients.

Characteristic	*n*	% of Total
Total patients enrolled	39	
Age at enrolment (years)		
Median (60)		
Range (30–84)		
Gender		
Male	16	59%
Female	23	41%
Stage of disease at baseline		
Stage IV		
M1a	5	13%
M1b	5	13%
M1c	19	49%
M1d	10	25%
Mutational status of tumour		
*BRAF* Mut	26	67%
V600E *	20	51%
V600K	5	13%
V600R	1	3%
*NRAS* Mut	3	7%
Q61K	2	5%
Q61L	1	2%
*BRAF/NRAS* WT	10	26%

* One case with an additional *BRAF S607F* mutation.

**Table 2 cancers-11-00157-t002:** Enriched cellular hallmarks in MCSP CTC fractions from metastatic melanoma cases.

Gene Set Name (Hallmark)	Number of Genes in Overlap	k/K	*p*-Value	FDR *q*-Value	Overlapped Genes
TNFA_SIGNALING_VIA_NFKB	20	0.1	3 × 10^−21^	1.5 × 10^−19^	*IL1B, NAMPT, NFKBIA, TNFAIP6, ABCA1, TNFAIP3, G0S2, TRIB1, NFIL3, SLC2A3, IRS2, SOCS3, GADD45B, SAT1, PLK2, BCL6, NR4A2, CCNL1, BCL2A1, B4GALT5*
INFLAMMATORY_RESPONSE	16	0.08	1.1 × 10^−15^	2.8 × 10^−14^	*IL1B, NAMPT, NFKBIA, TNFAIP6, ABCA1, RAF1, ADM, HIF1A, CSF3R, AQP9, IL18RAP, MEFV, SLC31A2, FAR2, KCNJ2, PROK2*
KRAS_SIGNALING_UP	9	0.045	3.3 × 10^−7^	5.5 × 10^−6^	*IL1B, TNFAIP3, G0S2, TRIB1, CSF2RA, ARG1, LY96, CLEC4A, TLR8*
COMPLEMENT	8	0.04	3.6 × 10^−6^	3.0 × 10^−5^	*TNFAIP3, RAF1, PLSCR1, CDA, CR1, F5, GCA, ITGAM*
HYPOXIA	8	0.04	3.6 × 10^−6^	3.0 × 10^−5^	*TNFAIP3, NFIL3, SLC2A3, ADM, IRS2, DDIT3, EXT1, FOXO3*
INTERFERON_GAMMA_RESPONSE	8	0.04	3.6 × 10^−6^	3.0 × 10^−5^	*NAMPT, NFKBIA, TNFAIP6, TNFAIP3, HIF1A, EIF4E3, PLSCR1, SOCS3*
IL2_STAT5_SIGNALING	7	0.035	3.5 × 10^−5^	2.2 × 10^−4^	*NFIL3, SLC2A3, GADD45B, PLSCR1, IL1R2, NFKBIZ, WLS*
MTORC1_SIGNALING	7	0.035	3.5 × 10^−5^	2.2 × 10^−4^	*NAMPT, NFIL3, SLC2A3, DDIT3, IDI1, FAM129A, IFRD1*
IL6_JAK_STAT3_SIGNALING	5	0.0575	4.6 × 10^−5^	2.6 × 10^−4^	*IL1B, SOCS3, CSF3R, CSF2RA, IL1R2*
BILE_ACID_METABOLISM	5	0.0446	1.5 × 10^−4^	7.7 × 10^−4^	*ABCA1, AQP9, IDI1, ACSL1, SULT1B1*

**Table 3 cancers-11-00157-t003:** Enriched cellular hallmarks in ABCB5 CTC fractions from metastatic melanoma cases.

Gene Set Name (Hallmark)	Number of Genes in Overlap	k/K	*p*-Value	FDR *q*-Value	Overlapped Genes
EPITHELIAL_MESENCHYMAL_TRANSITION	8	0.04	4.7 × 10^−7^	2.3 × 10^−5^	*VCAN, MYL9, PLOD2, LRP1, THBS1, FGF2, PTX3, CALU*
APICAL_JUNCTION	7	0.035	6 × 10^−6^	1 × 10^−6^	*VCAN, MYL9, INSIG1, ADAM9, SRC, VCL, RRAS*
HEME_METABOLISM	7	0.035	6 × 10^−6^	1 × 10^−6^	*H1F0, ENDOD1, BLVRA, CTNS, HEBP1, MARCH2, SNCA*
ANDROGEN_RESPONSE	4	0.04	4 × 10^−4^	4.5 × 10^−3^	*INSIG1, H1F0, GUCY1A3, ABCC4*
ESTROGEN_RESPONSE_EARLY	5	0.025	6.3 × 10^−4^	4.5 × 10^−3^	*ENDOD1, KLF10, MYOF, AMFR, ZNF185*
KRAS_SIGNALING_UP0	5	0.025	6.3 × 10^−4^	4.5 × 10^−3^	*GUCY1A3, TSPAN13, YRDC, EPB41L3, MMD*
MTORC1_SIGNALING	5	0.025	6.3 × 10^−4^	4.5 × 10^−3^	*PLOD2, INSIG1, DDIT3, STARD4, MLLT11*
COAGULATION	4	0.029	1.3 × 10^−3^	8.1 × 10^−3^	*LRP1, THBS1, ADAM9, SERPING1*
COMPLEMENT	4	0.02	4.9 × 10^−3^	2.7 × 10^−2^	*LRP1, ADAM9, SRC, SERPING1*
PI3K_AKT_MTOR_SIGNALING	3	0.029	5.6 × 10^−3^	2.7 × 10^−9^	*DDIT3, DUSP3, MAP2K6*

**Table 4 cancers-11-00157-t004:** Shared genes between CTC fractions and specifically expressed genes in single melanoma tumour cells.

CTC Fraction	Gene Symbols
MCSP CTC	*ST6GALNAC2, DDIT3, PLK2, LOXL4, CYSTM1, WIPI1, PYGL, GLUL, SAT1, SKP1, SEC31A*
ABCB5 CTC	*CYB5R1, H1F0, DUSP3, DDIT3, CREG1, ENDOD1, CALU, WIPI1, TXN2, SNCA*

## Data Availability

Raw and normalised data from gene expression microarray have been deposited into the freely publicly available NCBI′s Gene Expression Omnibus (GEO) database (accession number GSE113166). Website URL: https://www.ncbi.nlm.nih.gov/geo/query/acc.cgi?acc=GSE113166.

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
