# Peer review of "Immunomagnetic-Enriched Subpopulations of Melanoma Circulating Tumour Cells (CTCs) Exhibit Distinct Transcriptome Profiles"

_cancers, 2019, doi:10.3390/cancers11020157_

Reviewer 1 Report

The authors have addressed all the reviewers comments and the current version is publishable. 

Reviewer 2 Report

The authors did not provide sufficient response to my comments. However, given the importance of the work I have no objection to moving this paper forward 

Reviewer 3 Report

The responses and comments of the authors are acceptable and therefore, revised manuscript is appropriate for publication.

This manuscript is a resubmission of an earlier submission. The following is a list of the peer review reports and author responses from that submission.

Round  1

Reviewer 1 Report

The authors investigated two distinct melanoma CTC subpopulations enriched by the MCSP-coated and ABCB5-coated immunomagnetic beads, following by the gene expression profiling of MCSP-enriched and ABCB5-enriched CTCs. This study unveils the differences in their transcriptional programs and distinct biological pathways activated in these cells. However, the description in the discussion part should be more clarified before it is published. 

There are some questions in the main text: 

1) Page 3, Line 9: “ABCB5 was only co-detected with either TYR, PAX3, MLANA or MAGEA3 in 3 CTC fractions”, it is confusing, how could ABCB5 be co-detected with one of 4 type of genes in 3 CTC fraction? Does it mean that ABCB5 is co-detected with another two genes in 1 CTC fraction? 

2) Page 10, Line 37-41: It is confusing and unclear about the depiction. “WIPI1 is involved in melanosome formation and regulation of melanogenic enzymes, such as MITF and its targets, via negative regulation of TORC1 signalling”. According to Figure 3B, the down-regulated of mTORC1 will inhibit cell proliferation and survival. Does it mean that MIFT will inhibit cell proliferation and survival? But you also mentioned in the following description that “WIPI1 up-regulates MITF and MITF is involved in melanocyte differentiation, melanomagenesis and melanoma progression”. If MITF inhibit the cell proliferation and survival, why the up-regulation of MITF is involved in melanomagenesis and melanoma progression? Could you please make the association between WIPI1 and MITF clear?

3) Page 10, Line 43-44: “However, significant expression of genes belonging to the transcriptional program of MITF was not found in either CTC fraction”. What message does the author want to convey to us? What’s the association of this sentence with the depiction of “The limited overlap between gene expression profiles of single tumour cells and both CTC fractions might reflect differences in the transcriptional programming of CTCs and tumours” in this paragraph.

4) Page 9, Line 18: Typo error of “molecFular”, should be “molecular”. 

5) Page 13, Line 9: Typo error of “haves”, should be “have”.

Author Response

Reviewer 1

1. Page 3, Line 9: “ABCB5 was only co-detected with either TYR, PAX3, MLANA or MAGEA3 in 3 CTC fractions”, it is confusing, how could ABCB5 be co-detected with one of 4 type of genes in 3 CTC fraction? Does it mean that ABCB5 is co-detected with another two genes in 1 CTC fraction? 

Response: To clarify the statement we have modified the text in page 3 line 8:

On the other hand, the expression of ABCB5 was most commonly detected in ABCB5-enriched CTC fractions, but no other genes were co-detected with ABCB5. Conversely, in MCSP-enriched CTC fractions, ABCB5 transcripts were co-detected with TYR, PAX3, MLANA or MAGEA3 transcripts in 3 CTC fractions, whilst 5 CTC fractions had exclusive ABCB5 expression and the remaining expressed other melanocyte lineage genes (Figure 1)

2. Page 10, Line 37-41: It is confusing and unclear about the depiction. “WIPI1 is involved in melanosome formation and regulation of melanogenic enzymes, such as MITF and its targets, via negative regulation of TORC1 signalling”. According to Figure 3B, the down-regulated of mTORC1 will inhibit cell proliferation and survival. Does it mean that MIFT will inhibit cell proliferation and survival? But you also mentioned in the following description that “WIPI1 up-regulates MITF and MITF is involved in melanocyte differentiation, melanomagenesis and melanoma progression”. If MITF inhibit the cell proliferation and survival, why the up-regulation of MITF is involved in melanomagenesis and melanoma progression? Could you please make the association between WIPI1 and MITF clear?

Response: We found an enrichment of genes involved in mTORC1 signalling in both CTC fractions. Additionally, we detected WIPI1 upregulation in both fractions.    According to Ho et al., 2011, WIPI1 acts as one of many regulators of MITF expression and consequently its downstream targets, through WIPI1-mediated inhibition of mTORC1 signalling. However, our finding of WIPI1 upregulation as well as increased mTORC1 signalling does not fit with this model. We observed upregulation of genes involved in mTORC1 signalling, which has been associated with invasiveness in melanoma; signalling that should have been inhibited in melanocytes. In addition, we did not detect MITF upregulation or MITF transcriptional programming. Altogether, this confirms the dysregulation of signalling pathways during melanomagenesis and highlights the need for further study on the role of WIPI1 and mTORC1 in melanoma CTC biology. Based on the above and the reviewer’s comment, we have modified the following text (page 10 line 33):

In this study, we observed that MCSP and ABCB5 CTC fractions had 11 and 10 genes commonly expressed with single melanoma tumour cells, respectively. The limited overlap between gene expression profiles of single tumour cells and both CTC fractions might reflect differences in the transcriptional programming of CTCs and tumours. Interestingly, only WIPI1 and DDIT3 were shared between both CTC fractions and single melanoma cells. WIPI1 has been involved in melanosome formation and regulation of melanogenic enzymes, such as MITF and its targets, via negative regulation of TORC1 signalling [48]. Also, WIPI1 was identified as the signature gene able to differentiate primary melanomas from normal skin [49]. However, we observed upregulation of WIPI1 as well as of genes involved in mTORC1 signalling in both fractions, with no evidence of MITF transcriptional programing. Altogether, this suggests dysregulation of these signalling pathways in melanoma CTCs. Thus, further investigation of the role of WIPI1 and mTORC1 upregulation and signalling in CTC biology is required. On the other hand, DDIT3, a mediator of endoplasmic reticulum stress-mediated apoptosis [51], has not been implicated in melanoma biology to date. However, DDIT3 was up-regulated as result of the apoptotic activity triggered by the compound HA15 in melanoma cell lines [52] as was SEC31A [52], a gene found up-regulated in MCSP CTC fractions, Altogether, the role of WIPI1 and DDIT3 may indicate common molecular mechanisms in CTC biology. It is critical to understand such mechanisms in order to determine the changes in transcriptional programming that tumour cells undergo to become CTCs.

3. Page 10, Line 43-44: “However, significant expression of genes belonging to the transcriptional program of MITF was not found in either CTC fraction”. What message does the author want to convey to us? What’s the association of this sentence with the depiction of “The limited overlap between gene expression profiles of single tumour cells and both CTC fractions might reflect differences in the transcriptional programming of CTCs and tumours” in this paragraph.

Response: The statement was placed to relate to the discordance between our results and Ho et al. study linking WIPI1 upregulation to mTORC1 inhibition and MITF signalling. We have rearranged the paragraph in page 10 line 33, as part of the answer comment 2, above, to clarify this point.

4. Page 9, Line 18: Typo error of “molecFular”, should be “molecular”. 

Typo has been fixed in the document, as follows: This study confirms the phenotypic and molecular heterogeneity…

5. Page 13, Line 9: Typo error of “haves”, should be “have”.

Typo has been fixed in the document, as follows:  Raw and normalised data from the gene expression microarray have been deposited…

Reviewer 2 Report

Studying CTCs are very important to address questions about metastasis of tumor cells. It is also a good way to monitor tumor progression and response to therapies. 

It would be good if you could consider below points in your research:

1) To better address the heterogeneity of CTCs, it would be highly informative to do single CTC RT-qPCR analysis. This would help to further identify heterogeneity of CTCs within each sample of MCSP- and ABCB5-enriched CTCs.

2) Analyzing CTCs from same patients at 2-3 different time points to determine changes in the molecular signatures of isolated CTCs.

3) Compare your antibody-based immunomagnetic method with a label-free (no-antibody) method to identify more diverse types of CTCs with unique molecular characteristic other than expressing MCSP or ABCB5.

Author Response

Reviewer 2

1. To better address the heterogeneity of CTCs, it would be highly informative to do single CTC RT-qPCR analysis. This would help to further identify heterogeneity of CTCs within each sample of MCSP- and ABCB5-enriched CTCs.

Response: We thank the reviewer for his/her encouraging comments and we agree that a single cell qRT-PCR approach would definitely provide more informative and conclusive evidence of such heterogeneity than performing pooled RNA profiling of CTCs. However, the extreme rarity of CTCs (1-10 CTCs in 8 mL of blood), the significant technical challenge to isolate CTCs without contaminating white blood cells and the intrinsic heterogeneity of CTCs make downstream molecular interrogation of single CTCs extremely difficult Currently, we are individually picking CTCs from melanoma patients (e.g. Beasley et al., JCO Precision Oncology 2018) for further genome interrogation, but such methods are not ready for RNA profiling just yet. Additional, single cell strategies are currently being developed to unveil CTC heterogeneity.

Based on the reviewer’s comment, we have made the following insertion in the text to highlight this:

Page 11 line 13: Unbiased isolation and detection methods are required in order to capture the true heterogeneity of CTCs. Such studies will provide CTC subtype specific markers, other than MCSP and ABCB5 and will provide a better understanding of the biology of specific CTC subtypes as well as their association with melanoma biology. In addition, analysis of pure CTC fractions or of single CTCs is needed to ascertain their transcriptional programming without being confounded by the background of white blood cells.

2. Analyzing CTCs from same patients at 2-3 different time points to determine changes in the molecular signatures of isolated CTCs.

Response: We indeed agree with the reviewer that once we acquire the basal CTC heterogeneity before treatment, we can then see changes in the molecular signatures as the treatment progresses. This is where our research is currently heading.

3. Compare your antibody-based immunomagnetic method with a label-free (no-antibody) method to identify more diverse types of CTCs with unique molecular characteristic other than expressing MCSP or ABCB5.

Response: We have recently implemented label free microfluidic enrichment in order to interrogate the transcriptome of unbiasedly enriched CTCs (Aya-Bonilla et al., Oncotarget 2017).  In the future we also aim to examine an extended gene panel to further examine the phenotypic and molecular heterogeneity of melanoma CTCs.       

Reviewer 3 Report

The authors profiled the gene expression of CTC subpopulations immunomagnetic-captured by targeting either the melanoma-associated marker, MCSP, or the melanoma-initiating marker, ABCB5. They reported that these are two distinct melanoma CTC subpopulations with the ABCB5 subpopulation is enriched for genes involved in EMT. The study has high clinical translational value and is well designed and well presented. The reviewer only has some minor comments/suggestions:

1) Are MCSP and ABCB5 only expressed on CTCs? If not, the purification process needs to elaborate on the technical details and/or limitations? The reviewer suggests the author provide some representative images on the CTCs isolated from the patients' blood. It has been reported that CTCs exhibit a larger cell size compared to other cells in the circulation. 

2) Please comment whether the MACS process would have changed the gene expressions of CTCs?

3) It has also been reported that the CTC clusters are more dangerous than those single CTCs. Please comment on whether CTC clusters are detected in the present study

4) The field has advanced to the uses of single cell RNA seq to replace bulk PCR for cell phenotyping. Please acknowledge/discuss such considerations. 

Author Response

Reviewer 3

1. Are MCSP and ABCB5 only expressed on CTCs? If not, the purification process needs to elaborate on the technical details and/or limitations? The reviewer suggests the author provide some representative images on the CTCs isolated from the patients' blood. It has been reported that CTCs exhibit a larger cell size compared to other cells in the circulation. 

Response: As for most CTC isolation protocols, abundant WBC will remained in the CTC enriched fractions. Thus, we used blood obtained from healthy controls to determine the specificity of our assay as stated in page 4 line 12: “No expression of these 5 genes was detected in MCSP or ABCB5-enriched fractions derived from similarly evaluated healthy donors (N=16).” In addition, CTC fractions from healthy controls were used the whole genome expression microarray analysis.

Based on our own research and from other CTC groups (Gray et al., 2015; Aya-Bonilla et al., 2017; Khoja et al., 2014; Luo et al., 2014; Ramsköld et al., 2012), the melanoma CTCs seem to have a more differentiated phenotype, characterized by the expression of melanocyte lineage markers (e.g MCSP) or a melanoma-initiating phenotype, characterized by the expression of ABCB5. Despite this, as pointed out in the text, ABCB5 CTC fractions had melanocyte-origin transcripts and MCSP CTC fractions displayed ABCB5 expression. To underscore this point, we have inserted the following text in the document:

Page 11 line 14. Such studies will provide CTC subtype specific markers, other than MCSP and ABCB5 and will provide a better understanding of the biology of specific CTC types as well as determine their association with melanoma biology.

In regard to displaying some CTC images in the manuscript, unfortunately for this study the blood samples were used for CTC immunomagnetic enrichment followed by RNA extraction. Limited volume of blood was provided for study, and we could not spare an extra tube for immunodetection.  

2. Please comment whether the MACS process would have changed the gene expressions of CTCs?

Response:  We agree with the reviewer’s observation on changes in gene expression during the isolation process. Gene expression is a cellular mechanism that is responsive to physical and chemical changes in the surroundings of the cells. Thus, taking out cells from their natural environment unavoidably induces a change in gene expression. Although these changes are widely discussed in the literature, the methods employed reduced alterations in gene expression to a minimum by reducing the steps and time spent in processing the cells so that the gene expression observed reflects in its majority the original gene expression. In particular, blood samples were maintained at 4oC immediately after blood draw, were processed as fresh as possible, CTC fractions were kept in ice and RNA isolation was performed immediately after magnetic enrichment, similarly to other reported studies (ref 14, 16 and 37).

3. It has also been reported that the CTC clusters are more dangerous than those single CTCs. Please comment on whether CTC clusters are detected in the present study.

Response: We agree with the reviewer’s comment on CTC clusters. However, since we did not perform immunostaining on a matching blood sample due to the limited sample volume, it is unclear whether the gene signatures within the studied CTC fractions are derived from CTC clusters.

4. The field has advanced to the uses of single cell RNA seq to replace bulk PCR for cell phenotyping. Please acknowledge/discuss such considerations.

Response:  We agree that a single cell, either PCR-based or RNAseq, approach would definitely provide more informative and conclusive evidence of such heterogeneity than performing pooled RNA profiling of CTCs. However, the extreme rarity of CTCs  (1-10 CTCs in 8 mL of blood), the significant technical challenges to isolate CTCs without contaminating white blood cells and the intrinsic heterogeneity of CTCs make downstream molecular interrogation of single CTCs difficult. Single cell RNAseq strategies are currently being developed within our group in order to unveil the real heterogeneity of melanoma CTCs.

Based on the reviewer’s comment, we have made the following insertion in the text to highlight this point:

Page 11 line 14. Unbiased isolation and detection methods are required in order to capture the true heterogeneity of CTCs. Such studies will provide CTC subtype specific markers, other than MCSP and ABCB5 and will provide a better understanding of the biology of specific CTC subtypes as well as determine their association with melanoma biology. In addition, analysis of pure CTCs fractions or of single CTCs is needed to ascertain their transcriptional programming without being confounded by the background of white blood cells.